# Amyotrophic Lateral Sclerosis and Frontotemporal Lobar Degenerations: Similarities in Genetic Background

**DOI:** 10.3390/diagnostics11030509

**Published:** 2021-03-13

**Authors:** Eva Parobkova, Radoslav Matej

**Affiliations:** 1Department of Pathology and Molecular Medicine, Third Faculty of Medicine, Charles University and Thomayer University Hospital, 14059 Prague, Czech Republic; eva.parobkova@ftn.cz; 2Department of Pathology, First Faculty of Medicine, Charles University, and General University Hospital, 14059 Prague, Czech Republic; 3Department of Pathology, Third Faculty of Medicine, Charles University, and University Hospital Kralovske Vinohrady, 14059 Prague, Czech Republic

**Keywords:** amyotrophic lateral sclerosis, frontotemporal dementia, genetics, neuropathology

## Abstract

Amyotrophic lateral sclerosis (ALS) is a devastating, uniformly lethal progressive degenerative disorder of motor neurons that overlaps with frontotemporal lobar degeneration (FTLD) clinically, morphologically, and genetically. Although many distinct mutations in various genes are known to cause amyotrophic lateral sclerosis, it remains poorly understood how they selectively impact motor neuron biology and whether they converge on common pathways to cause neuronal degeneration. Many of the gene mutations are in proteins that share similar functions. They can be grouped into those associated with cell axon dynamics and those associated with cellular phagocytic machinery, namely protein aggregation and metabolism, apoptosis, and intracellular nucleic acid transport. Analysis of pathways implicated by mutant ALS genes has provided new insights into the pathogenesis of both familial forms of ALS (fALS) and sporadic forms (sALS), although, regrettably, this has not yet yielded definitive treatments. Many genes play an important role, with *TARDBP*, *SQSTM1*, *VCP*, *FUS*, *TBK1*, *CHCHD10*, and most importantly, *C9orf72* being critical genetic players in these neurological disorders. In this mini-review, we will focus on the molecular mechanisms of these two diseases.

## 1. Introduction

Amyotrophic lateral sclerosis (ALS) is a devastating, uniformly lethal progressive degenerative disorder of motor neurons that overlaps with frontotemporal lobar degeneration (FTLD) [1]. ALS is a motor neuron disorder (MND), lateral sclerosis, and spinal muscular atrophies [2]. ALS has a worldwide incidence of about two per 100,000 per year, and a prevalence of about five per 100,000 [3]. Clinical symptoms include weakness of the bulbar and limb muscles, hyperreflexia, spasticity of the arms and legs, and respiratory failure. These symptoms most commonly develop between 40 and 70 years of age, although a wider age range has been reported [4]. Although most cases of ALS are sporadic (sALS), ≈10% are familial (fALS), with predominantly autosomal dominant transmission [5]. Sporadic ALS accounts for most ALS cases, although genetic causes are also known to play a role [6]. fALS is linked to mutations at a specific genetic locus [7]. The clinical and pathological presentation of fALS and sALS are similar [8]. Many sALS studies (GWAS) have identified genes associated with ALS disease [9]. ALS is closely related to frontotemporal lobar degeneration (FTLD) [10]. Many studies have shown clinical, pathological, and genetic commonalities between them. Therefore, they are now considered two manifestations of one disease continuum, i.e., the ALS-FTD spectrum or the FTD-ALS spectrum with the broader name, i.e., frontotemporal lobar degeneration, being associated with motor neuron disorders (FTLD-MND) [11]. FTD-ALS patients have a poor prognosis with a mean survival of 2–3 years from the first onset of symptoms [12,13,14]. The reported heritability of FTLD-ALS is high: Approximately 50% of cases are considered familial [15,16]. ALS shares a common molecular etiology with frontotemporal dementia (FTD) [17], and approximately 15% of FTD patients develop motor neuron disease (MND), and conversely, up to 50% of MND show either direct signs of cognitive impairment [18,19] or minimal-mild disturbances in executive function [20,21]. These diseases are the extremes on a spectrum of clinically, pathologically, and genetically overlapping disorders [22], which suggests an overlap of disease mechanisms [23].

## 2. Mechanism and Molecular Pathology of ALS and FTLD

FTLD and ALS are two related neurodegenerative diseases forming the two ends of a disease spectrum. Behavioral FTLD is a form of dementia clinically characterized by progressive changes in behavior, personality, and language skills [24]. ALS involves the premature loss of upper and lower motor neurons, and a numeric decline in these neurons in the spinal cord, brainstem, and motor cortex [25], which leads to muscle weakness and atrophy [26]. The large number of genes and cellular processes associated with ALS has led to the suggestion that many disease mechanisms are involved. These include disturbances in RNA metabolism, impaired protein homeostasis, nucleocytoplasmic transport defects, impaired DNA repair, excitotoxicity, mitochondrial dysfunction, oxidative stress, axonal transport disruption, neuroinflammation, oligodendrocyte dysfunction, and vesicular transport defects [27]. While, traditionally, FTLD and ALS were considered to be two separate disease identities, it is now thought that FTLD and ALS form one clinical continuum, in which pure forms are linked by overlapping syndromes called FTLD-MND [17]. Recent advances in neuropathology and molecular genetics have started to reveal the biological basis for this clinical overlap. However, recent neuropathological findings suggest that FTLD cases present with distinct TDP-43 pathologies compared with ALS and FTLD-MND, indicating divergent disease pathogenesis mechanisms that nonetheless involve the same TDP-43 protein [28]. Although the underlying TDP-43 pathology does not always correlate with the genetics or disease phenotype, mutations of the progranulin (*GRN*) gene are generally associated with type A TDP-43 pathology, whereas hexanucleotide repeat expansions in *C9orf72*, which is the most common genetic cause of ALS and FTLD-TDP, most frequently result in the type B pathology [29,30]. Although *TARDBP* (the gene coding protein TDP-43) mutations are rare in ALS and FTD (<1%), the pathological aggregation of TAR DNA-binding protein 43 in affected brain regions and motor neurons is characteristic of the majority of ALS and FTD patients [11,31].

### 2.1. Protein and RNA Aggregates

The presence of protein and RNA aggregates in the cytoplasm of motor neurons is the hallmark of ALS [32]. The histopathological characteristics of ALS and FTD include abnormal accumulation of dysfunctional protein aggregations in the affected parts of the nervous systems [17]. Inclusions in ALS and FTD suggest that they share common pathogenic mechanisms leading to neurodegeneration and aggregation of specific inclusion proteins [33]. Similar ubiquitin-positive inclusions were observed in degenerating motor neurons of ALS patients [34], and aggregations of FUS protein have been reported in rare cases of ALS [35,36]. The most common protein inclusion in ALS is TDP-43 (encoded by the *TARDBP* gene) [32]; however, TDP-43 deposits have been observed in other neurodegenerative diseases such as Alzheimer’s disease (A.D.) [37,38], Lewy bodies dementia (DLB) [39,40], and corticobasal degeneration (CBD) [38,41]. The relevance of the concomitant TDP-43 pathology remains unclear, and attempts to correlate a concurrent TDP-43 pathology with clinical phenotypes have provided mixed results [38,42]. *C9orf72* (chromosome 9 open reading frame 72) has an essential role in stress granule formation, microglial function, and autophagy [43,44,45]. The accumulation of (GGGGCC) might lead to sequestration of RNA binding proteins and disruption of the translation of diverse mRNA or increased nucleolar stress [46]. These abnormal protein aggregates are thought to be the mechanism by which *C9orf72* expanded repeats contribute to ALS. One of the most frequently found proteins in neuropathological lesions is ubiquitin-binding protein p62 (sequestosome 1). Many experimental and clinical studies have shown that p62 plays a significant role in autophagy, an evolutionarily conserved pathway for the degradation of long-lived proteins and organelles. Dysfunction of the autophagy pathway may contribute to the pathology of various neurodegenerative disorders characterized by abnormal protein accumulation [47,48]. Recently, p62 was also shown to deliver ubiquitinated proteins, such as tau, and other crucial proteins involved in neurodegeneration, to proteasomes for degradation [49]. The build-up of p62-positive inclusions suggests defects in protein clearance pathways. Finally, a new role for p62 in maintaining mitochondrial integrity has recently been described [50]. A portion of p62 directly localizes within the mitochondria and supports stable electron transport by forming heterogeneous protein complexes [51]. Mutations in Superoxide Dismutase 1 (*SOD1*) produce an unstable protein deposited in the cytoplasm; oligomerization of unstable *SOD1* leads to aggregate formation [52]. Fused in sarcoma (*FUS*) is another RNA binding protein in which mutations can result in cytoplasmic aggregates. It is also a component of stress granules and may form p62 and TDP-43 positive aggregates [53]. It has been postulated that impaired autophagy could contribute to the accumulation of cytoplasmic aggregates [54]. Mutations in several autophagy genes have been associated with ALS, including Sequestosome 1 (*SQSTM1*), *SOD1*, optineurin (*OPTN*), valosin-containing protein (*VCP*), ubiquitin-2 (*UBQLN2*), and TANK-binding kinase 1 (*TBK1*) [55]. The removal of misfolded or damaged protein is critical for optimal cell functioning. In both the cytosol and the nucleus, major proteolytic pathways exist to recycle misfolded or damaged proteins, i.e., the UPS (ubiquitin proteasome system) and endosomal-lysosomal system (ELS) [56]. An impaired UPS is thought to be associated with the formation of proteinaceous inclusions in many neurodegenerative disorders [57].

### 2.2. Mitochondrial Dysfunction

Mitochondria dysfunction is at least partly responsible for the broad clinical spectrum of ALS and FTD. Mitochondrial dysfunction has been implicated in ALS motor neuron death [58]. Fragmentation of mitochondria and changes in mitochondrial morphology and expression of fusion/fission proteins are well described in ALS and have pronounced effects on normal mitochondrial function [59]. Mitochondria from ALS patients have impaired Ca^2+^ homeostasis and increased production of reactive oxygen species (ROS). ROS are associated with oxidative-related damage, including changes in protein carbonylates and tyrosine nitration [60]. Mitochondria are essential for cellular respiration, calcium buffering, and apoptosis. Neurons are particularly sensitive to mitochondrial dysfunction given their high metabolic rate [61]; as such, the presence of abnormal or dysfunctional mitochondria in neurons is thought to be a contributing factor in ALS. Mitochondria are of particular importance in neurons. Neurons have high metabolic requirements, with the brain consuming 20% of the body’s resting ATP production despite being only 2% of the body’s mass [62,63]. Moreover, mitochondria are essential calcium buffering organelles in neurons and function to modulate local calcium dynamics, for example, modulating neurotransmitter release [64]. Many proteins that have been linked to familial and sporadic ALS, including *SOD1*, *TDP-43*, *FUS*, and *C9orf72*, show interactions with mitochondria [65,66,67]. Direct evidence that disruption of mitochondrial structure (and as a result disruption of mitochondrial function; see below) may contribute to the etiology of ALS comes from the discovery of causative mutations in the mitochondrial protein Coiled-coil-helix-coiled-coil-helix domain-containing protein 10 (CHCHD10); this protein is localized to contact sites between the inner and outer mitochondrial membrane [68]. ALS-associated mutations in *CHCHD10* can disrupt mitochondrial cristae and profoundly affect the mitochondrial structure [69]. Mitochondrial DNA instability disorders are responsible for frontotemporal dementia [68]. In recent years, a growing list of FTD genes responsible for mitochondrial DNA instability has been reported [70]. The c.176C>T mutation in the *CHCHD10* gene was described in an FTD-ALS patient whose family was originally from Catalonia (Spain), with affected individuals carrying a missense mutation in the *CHCHD10* gene. Functional characterization of the *CHCHD10* mutant identified in the family showed fragmentation of the mitochondrial network and the loss of cristae junctions [68]. *CHCHD10* is a novel gene responsible for the clinical spectrum of ALS-FTD, which raises the intriguing prospect of an underlying mitochondrial basis for this group of disorders.

### 2.3. Impaired DNA Repair

Impair DNA repair is another suggested mechanism that may contribute to ALS pathogenesis. Two of the best-studied ALS-linked proteins, TDP-43 and FUS, function to prevent or repair transcription-associated DNA damage [71]. FUS, in particular, seems to play an essential role in this regard and is involved in the repair of double-stranded DNA breaks via both homologous recombination and non-homologous end-joining repair mechanisms [72,73]. Variations in the genes of other ALS-linked RNA-binding proteins, including TATA-box binding protein associated factor 15 (*TAF15*), senataxin (*SETX*), and RNA-binding protein EWS (*EWSR1*), have also been linked to impaired DNA damage repair, further implicating the breakdown of this process in ALS pathogenesis [74,75,76].

### 2.4. Axonal Transport Defects

Axonal transport defects are a common observation in various neurodegenerative diseases, and mutations in components of the axonal transport machinery have unequivocally shown that impaired axonal transport can cause neurodegeneration [77]. The underlying cause of axonal transport defects in ALS is not fully understood [78]. Several mechanisms by which axonal transport may be perturbed in sporadic ALS and familial ALS by mutations in non-axonal transport genes have been proposed based mainly on studies of mutant *SOD1*-related ALS [79]. These include reductions in [1] microtubule stability, [2] mitochondrial damage, [3] pathogenic signaling (which alters phosphorylation of molecular motors and thereby regulate their function or through phosphorylation of cargo, such as neurofilaments, to disrupt their association with motors), and [4] protein aggregation [79,80]. There is evidence that TDP-43 may also be involved in the axonal transport defects seen in ALS [27]; axonal transport defects are commonly seen in neurodegenerative diseases [78]. TDP-43 has an important role in regulating axonal growth and impairment in posttranscriptional regulation of mRNAs in the cytoplasm of motor neurons [81]. Mutations in the genes coding for axonal transport first came to light when LaMonte et al. showed that disruption of the dynein/dynactin interaction by postnatal overexpression of p50/dynamitin, a 50-kDa subunit of dynactin encoded by *DCTN2*, caused reduced axonal transport in motor neurons and consequently led to a late-onset progressive motor neuron disease phenotype in the transgenic mice [82]. This study was followed by several studies showing that loss-of-function mutations in *DCTN1* cause a slowly progressive autosomal dominant distal hereditary motor neuropathy with vocal paresis (HMN7B) and ALS [79,83,84]. One key role for dynein in the neuron may be the removal of misfolded or degraded proteins from the cell periphery and the transport of these proteins back to the cell body for degradation. Dynein is also involved in the accumulation of misfolded proteins into aggresomes [78]. The next major family of microtubule-based molecular motors is kinesin. Kinesin moves mostly toward the plus end of microtubules, and was the first axonal transport motor to be identified. Now known as kinesin-1, it is a unidirectional motor, driving plus end-directed motility along microtubules in vitro [85]. Microtubules (e.g., α-tubulin) play a pivotal role in developing and maintaining neuronal cell structure and function, and they serve as essential tracks for both fast and slow long-distance axonal transport [86]. Several variants of the α-tubulin gene (i.e., *TUBA4A*), which destabilize the microtubule network and diminish its re-polymerization capability, have been identified as possible causes of ALS [87]. Whether these mutations affect axonal transport has not yet been determined, but since axonal transport prefers stable microtubules, they will likely have a detrimental effect [88].

### 2.5. Altered RNA Metabolism

As key regulators of RNA metabolism, RNA-binding proteins (RBP) play a critical role in maintaining the normal function of neuronal systems. RNA-binding proteins are involved in several aspects of RNA metabolism, including splicing, transcription, transport, translation, and storage in stress granules [89]. The aggregation of RBP is a pathological hallmark of amyotrophic lateral sclerosis and frontotemporal lobar degeneration. Interestingly, many of the ALS-linked RNA-binding proteins contain prion-like domains that are involved in stress granule formation or dynamics, including TDP-43, FUS, TAF15, ESWR1, hnRNPA1, and hnRNPA2B1 [90]. Mutations in ALS genes contribute to the etiology of FTD and vice versa [10]. Many ALS-causing mutations impact proteins involved in RNA metabolism, including RNA-binding proteins such as TDP-43, FUS, and heterogeneous nuclear ribonucleoprotein A1 (hnRNPA1) [91]. These and related RNA-binding proteins are components of organelles without membranes found in the nucleus (e.g., nuclear speckles and nucleoli) and cytoplasm (e.g., processing bodies and stress granules) in neurons and other cell types [92,93,94].

### 2.6. Mechanisms Leading to Dysregulation of RBP in ALS

Mutations in genes encoding many RBP are highly associated with ALS. In addition, dysregulation of RBP as a result of compromised nucleocytoplasmic trafficking, posttranslational modification (PTM), aggregation, and sequestration by abnormal RNAs also contribute significantly to disease pathogenesis. This section will briefly discuss the underlying mechanisms resulting in RBP dysregulation in ALS [95].

In response to a variety of stressors such as heat shock and oxidative insult, TDP-43 and FUS translocate from the nucleus and associate with cytoplasmic stress granules (SG), which are dense aggregations of protein-RNA complexes [96,97]. RBPs recruited to stress granules under conditions of chronic stress are capable of forming insoluble protein aggregates, even when other components of the stress granules have dissociated from the complex [98]. These granules facilitate cell survival by the translational arrest of non-essential transcripts and pro-apoptotic proteins when under stress [99]. Prion-like domains are thought to be vital for the reversible assembly of stress granules due to their capacity to form multiple transient weak interactions [90]. RBP also contains low complexity sequence domains (LCD), i.e., a glycine-rich domain that promotes protein aggregation [100] and contains RNA-recognition motifs (RRM) necessary for the nucleic acid binding functions of the protein [101]. Each protein also contains a nuclear localization sequence (NLS) that directs the subcellular localization of the protein to the nucleus under normal conditions [102]. Mutations in genes encoding NLS and LCD (Figure 1) lead to cytoplasmic retention and inclusion formation in cultured cells [103]. More than 250 proteins with aggregation-prone properties that are likely to contribute to neurodegeneration have been identified [104].

Low Complexity sequence Domains (LCD), Nuclear Localization Sequence (NLS). Mutations that occur in these domains (LCD and NLS) can trigger the same pathological cascade, which leads to a deterioration in the dynamics of stress granules (updated following original citation Baradaran-Heravi et al., 2019).

These studies suggest that such RBP could or should be considered as potential functional candidate genes in genetic studies. *RBM45*, an RNA-binding protein, is most likely a causal gene for ALS-FTD. In addition, a novel and evolutionary conserved structural element homo-oligomer assembly (HOA) domain has been identified. It is located within the linker region between RNA-recognition motifs (RRM2 and RRM3), which are essential for the self-association and oligomerization of *RBM45* (Figure 2). Since *RBM45* contains three RRM domains, it may associate with TDP-43 and FUS through RNA-protein interactions [105].

*RBM45* lacks the typical low complexity domain (LCD), which is actually common in RBPs; it has a suspicious homo oligomerization domain that, similar to LCD, mediates self-assembly through homo oligomerization and interaction with other proteins (updated following original citation Li et al. (2015).

Another RBP is *TIA-1* that promotes the assembly of stress granules discrete cytoplasmic inclusions into which stalled translation initiation complexes are dynamically recruited in cells subjected to environmental stress [106]. *TIA-1* is a modular protein composed of three RNA recognition motifs and a carboxy-terminal glutamine-rich motif that is structurally related to prion protein (PRD). Overexpressed *TIA-1* induces SG formation and represses reporter gene expression, whereas the isolated prion-related domain (PRD) of *TIA-1* forms cytoplasmic microaggregates [107]. These data suggest that the PRD is capable of self-oligomerization, just like *RBM45*.

### 2.7. Neuroinflammation

The inflammatory environment associated with ALS changes with disease progression and involves both neurotoxic and neuroprotective aspects. Neuroinflammation associated with neuronal loss is characterized by microglia and astrocyte activation, overproduction of inflammatory cytokines, and infiltration of T lymphocytes [108]. The secretion of inflammatory proteins by activated microglia leads to the potentially neurotoxic activation of astrocytes, which may contribute to the death of neurons and oligodendrocytes [109]. Genes that influence these functions are highly expressed in microglia and include *C9orf72*, *TBK1*, *OPTN*, *SQSTM1*, and *PGRN* [27,110,111,112]. Recent preclinical studies suggest that dysfunction of the gastrointestinal tract may also play a role in ALS pathogenesis by modifying the gut microbiota-brain axis [113]. Microglia and astrocytes in the central nervous system were shown to be regulated by metabolites derived from symbiotic gut microbes; the pathway inhibited neuroinflammation and neurodegeneration in an experimental autoimmune encephalomyelitis model [114,115]. Generation of low molecular weight metabolites by the gut microbiome is one postulated mechanism; these compounds are capable of passing through the blood-brain barrier and influencing neuronal function [116,117,118]. A correlation between ALS and altered composition of the gut microbiota was previously tested in animal models [117,119]. Several preliminary studies have analyzed the fecal microbiota in ALS patients, but with no conclusive results [120,121]. Whether the gut microbiome influences ALS is still controversial and remains a matter of debate.

## 3. Overview of ALS and FTD Genes

The advent of next-generation sequencing technologies, such as whole-genome sequencing (WGS) and whole-exome sequencing (WES), has led to a wave of novel genes associated with ALS [122,123]. A recent study found that multiple genetic variants can interact simultaneously to increase ALS susceptibility; these oligogenic cases of ALS may not appear familial in a conventional Mendelian sense; nonetheless, they may underlie the sporadic form of the disease [124,125,126]. More than forty-six ALS-related genes overlap with ALS genes linked to hereditary spastic paraplegia (HSP), FTD, mitochondrial disease, and lower motor neuropathies (LMN) [127] (Figure 3). Most of the heritability of FTD is accounted for by autosomal dominant mutations in three genes: Progranulin (*GRN*), microtubule-associated protein tau (*MAPT*), and *C9orf72* [128]. In recent years, an increasing number of mutations in other genes have been associated with autosomal dominant FTD, e.g., *VCP* (2004), *CHMP2B* (2005), *TARDBP* (2008), *FUS* (2009), *SQSTM1* (2012), *CHCHD10* (2014), *TBK1* (2015), *OPTN* (2015), *CCNF* (2016), and *TIA1* (2017). Recent studies have identified TBK1 as probably the fourth most common genetic cause overall of FTD, accounting for between 1% and 2% of all cases (although the pathogenic nature of many of the reported missense variants remains unclear) [129]. The first ALS gene, cytosolic superoxide dismutase (*SOD1*), was reported in 1993 [130] as well as other genes such as TAR DNA binding protein (*TARDBP*) [131,132,133,134], angiogenin (*ANG*) [135], fused in sarcoma (*FUS*) [35,36], optineurin (*OPTN*) [136], and the recently described chromosome 9 open reading frame 72 (*C9orf72*) [26,137]. An overview of recently proposed ALS genes that were identified based on rare genetic variants (*TBK1*, *CHCHD10*, *TUBA4A*, *CCNF*, *MATR3*, *NEK1*, *C21orf2*, *ANXA11*, *TIA1*) and their potential relevance to the genetic etiology of frontotemporal dementia have also been described [10].

### 3.1. SOD1

Mutations in *SOD1* account for the second most common cause of fALS after *C9orf72* [122]. *SOD1* mutations account for 15–20% of fALS pedigrees [138,139] and, until the discovery of *C9orf72*, was the most commonly identified gene in ALS. In most families harboring *SOD1* gene mutations, disease penetrance is >90% by age 70 yrs [140], and more than 170 mutations have now been detected in the fALS *SOD1* gene [141]. This likely remains true in many non-white populations, where *C9orf72* is much less common. Although multiple hypotheses have been proposed to explain mutant *SOD1*-mediated toxicity [142], the exact mechanism(s) responsible for motor neuron degeneration remains unresolved. Mitochondrial dysfunction is thought to contribute to the pathogenesis [143], and a proportion of the predominantly cytosolic *SOD1* has been reported to localize to mitochondria under certain conditions [144,145,146]. Pickles et al. reported that wild-type *SOD1* proteins are only partially located in the mitochondria, while mutant proteins show an increased propensity to be located in mitochondria, suggesting mitochondria involvement in the ALS etiology [147]. Another key player that directly interacts with *SOD1* is voltage-dependent anion channel 1 (VDAC1), a mitochondrial channel protein. Translocation of ions and proteins between mitochondria and the cytoplasm may be affected by mitochondria-associated misfolded mutant *SOD1* [148], leading to increased mitochondria dynamic abnormalities and fragmentation [149]. Pure upper motor neuron (UMN) and lower motor neuron (LMN) forms have also been described, representing opposite clinical ends of the MND spectrum [9]. Mutations in the SOD1 gene could be associated with significant LMN involvement with or without signs of UMN. The A4V missense mutation occurs in around 40% of patients in North America, but is rare in the European population. LMN signs predominate, with features of UMN being mild or absent. Disease progression is particularly rapid, with a median survival of 1.2 years from disease onset [150]. The A4T mutation is also associated with a similarly rapid disease course for LMN predominant syndrome [151]. In contrast, the G93C mutation has been associated with a pure clinical phenotype of LMN, i.e., without bulbar involvement and a more favorable prognosis (i.e., a median survival of 153 months) [152].

### 3.2. TARDBP and FUS

One consistent pathologic finding in ALS and FTD is the presence of heavily ubiquitinated neuronal cytoplasmic inclusions, which in 2006 were found to contain TDP-4 [153]. TDP-43, an RNA-binding protein, is implicated in multiple aspects of RNA processing, including regulation of transcription, splicing, transport, and mRNA stabilization [154]. Significant modifications of TDP-43 have been identified as being hyperphosphorylated and proteolytically cleaved by caspases. Activation and cleavage of TDP-43 is a key molecular step linking cellular redistribution and toxicity to the neurodegeneration observed in TDP-43 proteinopathies [155]. TDP-43 has a promiscuous protein interaction pattern with more than 200 targets reported, suggesting an involvement in a vast array of intracellular events [156]. Abnormal molecular weight TDP-43 fragments have been observed in neurons and astrocytes in patients with a spectrum of neurodegenerative diseases, including 95% of familial and sporadic ALS [157], making it an interesting candidate for all forms of the disease. *TARDBP* mutations were initially identified [131] as a direct consequence of the identification of the TDP-43-derived protein species as the principal constituent of the aggregates found in the upper and lower motor neurons of ALS patients without *SOD1* mutations and in FTLD-UPS [158,159]. Whereas 5% of familial ALS patients have the *TARDBP* mutation, mutations are rarely found in FTLD and FTD-MND [132,160].

### 3.3. C9orf72

The protein encoded by *C9orf72* is mainly related to autophagy, endosomal transport, and immune function. According to statistics, about 40–50% of fALS and 10% of sALS patients carry the *C9orf72* expanded alleles. In one study, the *C9orf72* expansion accounted for 11.7% of familial FTD [26]. The pathogenic alleles of *C9orf72* may have hundreds or even thousands of GGGGCC hexanucleotide repeats. A large number of clinical studies have shown that about 700–1600 GGGGCC hexanucleotide repeats are inserted into the intron located between the two untranslated optional exons 1a and 1b of the *C9orf72* gene [127,161,162,163]. Disease penetrance of *C9orf72*-related ALS is thought to be nearly 100% by the age of 80 yrs. No predictions for individual phenotypes, i.e., ALS, FTD, or ALS/FTD, the exact age at onset, the disease course, and disease duration is currently possible [162].

## 4. Novel ALS Genes

### 4.1. KIF5A

Kinesins are microtubule-based motor proteins involved in the intracellular transport of organelles within eukaryotic cells. *KIF5* genes are expressed in neurons, and transcription products function to transport cargo by binding to distinct adaptor proteins [164]. The central role of kinesins in axonal transport leads us to speculate that mutations in *KIF5A* cause disease by disrupting axonal transport. *KIF5* is responsible for the axonal transport of neurofilaments [165], and *KIF5A* knockout mice display abnormal neurofilament transport [166]. Abnormal accumulation of neurofilaments is a pathological hallmark of ALS, and rare mutations in the neurofilament heavy polypeptide (NEFH) are associated with ALS [167]. *KIF5* also contributes to the transport of mitochondria [164], and the impaired mitochondrial transport and function is another common hallmark of ALS patients [168,169,170,171].

### 4.2. TBK1 and OPTN

The *TBK1* and *OPTN* genes encode functionally related proteins that recently gained increased attention from the ALS research community. TBK1 (tumor necrosis factor (TNF) receptor-associated factor NF-κB activator (TANK)-binding kinase 1), also known as NAK or T2K, recently attracted the attention of human geneticists, immunologists, and neurologists alike for its critical role in pathologies of the central nervous system (CNS). *TBK1* is involved in the activation of various cellular pathways leading to IFN and pro-inflammatory cytokine production following infection [172], autophagic degradation of protein aggregates or pathogens [173,174], and homeostatic cellular functions such as cell growth and proliferation [175]. The majority of *TBK1* mutations are loss-of-function (LOF), leading to loss of the mutant transcript through nonsense-mediated mRNA decay (NMD) [10]. *TBK1* LOF mutations account for 3–4% of ALS-FTD patients [10]. Nonsense and frameshift mutations cause major disruptions to *TBK1* and may decrease its expression at both the mRNA and protein level, implying that *TBK1* haploinsufficiency contributes to the development of ALS [111,176].

### 4.3. CHCHD10

Many additional Coiled-coil-helix-coiled-coil-helix domain-containing protein 10 (*CHCHD10*) variants are now known to cause ALS, FTD, and other related degenerative diseases [177,178]. However, the degree of their pathogenicity and penetrance is undetermined. Variants causing ALS have been described; however, experimental evidence does not support the assumption that all disease-causing variants have the same mode of action [179]. *CHCHD10* G58R and *CHCHD10* G66V were identified in mitochondrial myopathy and late-onset spinal muscular atrophy [180,181,182] such as *VCP* and Matrin 3 (*MATR3*), which also exhibit clinical pleiotropy, including myopathy [183,184]. *CHCHD10* expression in patient tissues is unaffected, and *CHCHD10* S59L overexpression causes mitochondrial defects similar to those in affected patients [68]. This suggests that *CHCHD10* S59L is a dominant gain-of-function mutation [185]. To date, no published study supports routine diagnostic or predictive testing for *CHCHD10* variants in pure ALS [178].

### 4.4. MATR3

Matrin 3 is a highly conserved, inner nuclear matrix protein with two zinc finger domains and two RNA recognition motifs (RRM). It has been proposed that it stabilizes certain messenger RNA species [186]. Another study suggests that *MATR3* regulates alternative splicing events by binding to introns flanking repressed exons [10,187]. *MATR3* mutations are observed in 0.5–2% of ALS patients [188,189,190], but no studies so far have identified any *MATR3* mutations in FTLD [10].

### 4.5. HNRNPA1 and HNRNPA2B1

Over the last decade, dysfunction of hnRNPs has become closely linked to neurodegenerative diseases, most prominently amyotrophic lateral sclerosis and frontotemporal dementia, two diseases with significant genetic and pathological overlap [91]. hnRNP A1 and hnRNP A2B1 share similar domain architectures and are primarily localized in the nucleus [191,192,193]. hnRNP A2B1 are components of RNA transport granules found in neurons [194]. In addition, hnRNP A1 and hnRNP A2B1 translocate to the cytoplasm in response to stress and are recruited to stress granules [195,196]. hnRNP A1 and hnRNP A2B1 are associated with <1% of familial and sporadic forms of ALS; instead, they are more frequently associated with the broader spectrum multisystem proteinopathy (MSP) disorder [192,197].

### 4.6. TIA1

As a gene that has only recently been associated with ALS, *TIA1* harbors several ALS- and ALS/FTD-associated mutations in the low-complexity sequence domain (LCD) [31]. Notably, the LCD of *TIA1* is also the site of a mutation that causes Welander distal myopathy, a myopathy characterized by TDP-43-positive inclusions [198] and p62 [199,200]. TIA1 assembles in organelles without membranes, e.g., stress granules [31].

### 4.7. NEK1 and C21orf2

*NEK1* encodes a member of the highly conserved NIMA (never in mitosis gene A) kinase family. It is a serine/threonine kinase involved in cell-cycle regulation, ciliogenesis, mitochondrial membrane permeability, and DNA damage repair [201,202,203]. Interestingly, in DNA damage repair, *NEK1* was shown to interact with *C21orf2*, which was recently associated with an increased ALS risk [204,205,206] (Table 1). Mutations in both *NEK1* and *C21orf2* are linked to skeletal disorders and axial spondylometaphyseal dysplasia [206]. Additionally, *C21orf2* may participate in the DNA repair process via interaction with *NEK1* [204]. Loss of function (LOF) variants account for about 1% of patients, and an interpretation of the pathogenicity and penetrance is complicated by the observation of occasional LOF variants in asymptomatic carriers [206]. In neurons, NEK proteins take part in maintaining the cytoskeleton network [207,208,209], which was previously linked to an ALS etiology via *TUBA4A* and *PFN1* [87,208,210]. With only 5–10% of sporadic ALS cases harboring disease-associated mutations in known ALS genes [211], the remainder of sALS cases are presumed to represent a complex disease process influenced by both genetic and environmental exposures. Efforts to identify genetic risk factors have largely focused on genome-wide, and candidate gene association studies have met with only occasional success. Variants of the *UNC13A* gene, for example, have been associated with susceptibility to ALS and shorter survival times [212]. Intermediate length trinucleotide repeat expansions of both the *ATXN1* and *ATXN2* genes also increase the risk of disease, particularly for *C9orf72* expansion carriers in the case of *ATXN1* [213,214]. A copy number variation in the *EPHA3* gene, in contrast, has been flagged as a potential protective factor for ALS [215].

## 5. Oligogenic/Polygenic Sporadic ALS

Variants in these genes are more likely to have a significant effect, and having the genotype greatly increases the probability of ALS; in other words, these variants show moderate to high penetrance; however, gene variants with low penetrance are also of interest, even though they only modestly increase the risk for any individual [216]. In at least some cases, ALS can be oligogenic, with affected individuals carrying more than one rare variant implicated in ALS [126,217]. An oligogenic basis of amyotrophic lateral sclerosis is debatable, since the presence of some variants have an uncertain significance [218]. However, a combination of a known pathogenic variant with one of uncertain significance is considered oligogenic inheritance by some [219]. What we do know is that oligogenic/polygenic sporadic ALS cases showed an earlier age of onset [220].

## 6. Future Questions

Our understanding of the biological and genetic basis of ALS, as well as our ability to care for ALS patients, has improved substantially over the last few years. Genetic causes of ALS have been identified in both sporadic and familial patients, and the number of disease-associated genes is still increasing [162]. One might assume that most of the monogenic forms have already been identified, but the issue of “heritability” is still unresolved, which might be explained by rare variants with large effect sizes [221]. The identification of genetic causes of ALS will help develop new therapeutic approaches, either through the identification of shared disease pathways such as the TDP-43 pathology or by targeted therapies for known mutations [222]. Currently, antisense oligonucleotide trials in SOD1- and *C9orf72*-related ALS are being conducted [223]. Besides riluzole, a second medication, edaravone, was approved by the FDA last year [224]. Whether it is beneficial to all ALS patients or just to subgroups needs to be evaluated over the next few years [2]. In the future, biomarkers will hopefully help monitor disease progression and genomics, and transcriptomics will help to further personalize treatment based on each patient’s individual disease subtype. For now, we need to define the molecular mechanisms that link specific disease-causing mutations to stress granule dysfunction and the accumulation of pathological inclusions.

## Figures and Tables

**Figure 1 diagnostics-11-00509-f001:**
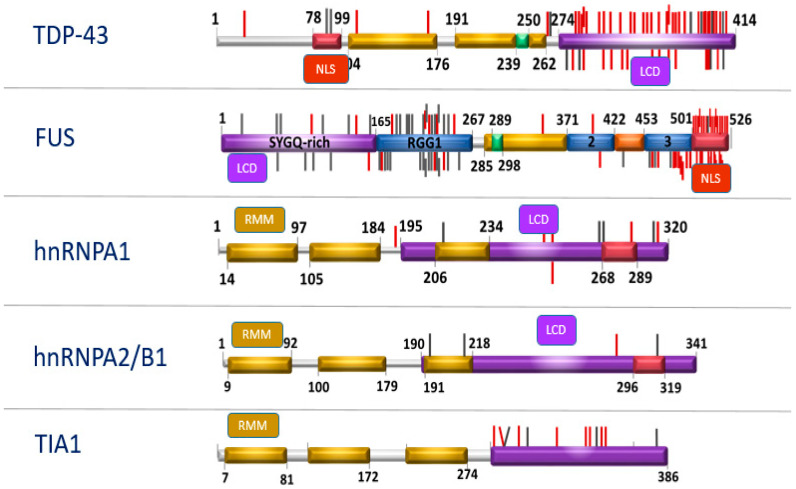
Structure of RNA-binding protein (RBP) genes.

**Figure 2 diagnostics-11-00509-f002:**
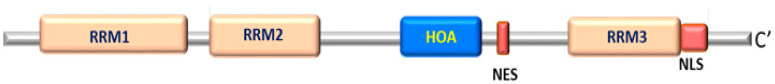
Structure of the RBM45 gene.

**Figure 3 diagnostics-11-00509-f003:**
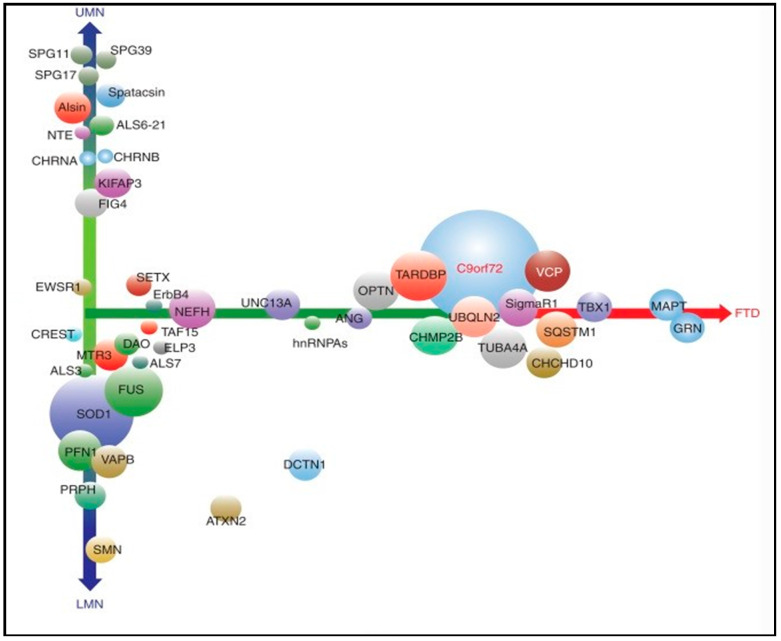
ALS-related genes. Symbols: LMN: Lower motor neuropathies, FTD: Frontotemporal dementia, UMN: Upper motor neurons. *X*-axis: Overlap with frontotemporal dementia, *Y*-axis: The extent to which corticospinal versus lower motor neurons are involved (updated following Ghasemi M. 2018).

**Table 1 diagnostics-11-00509-t001:** Overview of recent ALS genes in different form ALS and TDP.

Gene	Locus	Inheritance	Mutation Freguency						Protein Function
			fALS %	sALS %	Overal ALS %	fTDP %	sTDP %	Overal TDP %	
**TBK1**	12q14.2	AD	3	<1	1.3	2	1	<1	autophagy
**CHCHD10**	22q11.23	AD	2	<1	<1	<1	<1	<1	mitochondrial dysfunction, synaptic integrity
**TIA1**	2p13.3	AD	2		<1				RNA metabolism
**ANXA11**	10q22.3	AD	1.2	<1	1.1				apoptosis, exocytosis, cytokinesis
**CCNF**	16p13.3	AD	0.6–3.3	<1	<1			4	UPS
**NEK1**	4q33	?	1–2	<1					DNA damage, mitochondrial membrane regulation
**C21orf2**	21q22.3	?	<1	<1	<1				DNA damage
**MATR3**	5q31.2	AD	1	1	<1				RNA metabolism
**TUBA4A**	2q35	AD	1	<1	<1	<1		<1	cytoskeletal dynamics

AD, autosomal dominant; UPS, ubiquitun-proteasome system.

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
