# Peer review of "Amyotrophic Lateral Sclerosis and Frontotemporal Lobar Degenerations: Similarities in Genetic Background"

_diagnostics, 2021, doi:10.3390/diagnostics11030509_

Round 1

Reviewer 1 Report

The manuscript from Parobkova and Matej aims at reviewing the genetic and molecular overlap between ALS and FTD, two extreme presentations of a spectrum of disease defined as ALS-FTD. Although the topic is important and timely, the manuscript falls short of expectations. The structural organization of the review has major issues, such as the exact same content repeated many times. For example, the "The molecular genetics of ALS and FTLD-MND and the molecular biology of diverse mutant ALS genes." section overlaps extensively with many of the previous paragraph. The focus of the review on describing the overlap in the genetic background of ALS and FTD gets often lost and entire subsections do not even mention FTD. As an example, mutation in SOD1 are known to associate with pure motoneuronal disease, and yet ample space is dedicated to that. Similarly, in the “Mitochondria dysfunction” section FTD is not even considered. The section of RNA metabolism is very short, and the authors neglect to mention any of the many studies that have investigated RNA dysregulation in ALS and FTD using human tissue, animal, and cellular disease models. Only two broad-scoped reviews are referenced. Some statements are incorrect, such as the fact that C9ORF72 protein forms cytoplasmic aggregates (page 3, line 104), and citations imprecise (e.g. Ref#29 on line 179). Overall, a major rework of both the content and structure of this review would be required to make it acceptable for publication.

Author Response

Reviewer 1

  1. The structural organization of the review has major issues, such as the exact same content repeated many times. For example, the "The molecular genetics of ALS and FTLD-MND and the molecular biology of diverse mutant ALS genes." section overlaps extensively with many of the previous paragraph.

We agree with the reviewer; this point deserves considerable change.

  1. We renamed two section headings (1) Mechanism and molecular pathology of ALS and FTLD and (2) Overview of ALS and FTD genes.
  2. The overlapping text in the table of contents was removed.
  3. The text has been supplemented and corrected with new figures and a table.

Section: Mechanism and molecular pathology of ALS and FTLD. The text in this section has been revised.

Protein and RNA aggregates

The accumulation of (GGGGCC) might lead to sequestration of RNA binding proteins and disruption of the translation of diverse mRNA or increased nucleolar stress (46).

The removal of misfolded or damaged protein is critical for optimal cell functioning. In both the cytosol and the nucleus, major proteolytic pathways exist to recycle misfolded or damaged proteins, i.e., the UPS (ubiquitin-proteasome system) and endosomal-lysosomal system (ELS) (56). An impaired UPS is thought to be associated with the formation of proteinaceous inclusions in many neurodegenerative disorders (57).

Mitochondrial dysfunction: The same as comment 3.

New paragraph: Perturbations of cytoskeletal and axonal dynamics

Genetic analyses have identified several ALS-TDP genes that, as a group, identify perturbations of cytoskeletal and axonal dynamics as a central element in ALS pathogenesis (77). The most common are disorders of the genes profilin-1 (PFN1), dynactin (DCTN1), and tubulinA4A (TUBA4A), affecting cytoskeletal architectures and dynamic functions of distal axons and dendrites (78).

Altered RNA Metabolism: The same as comment 4.

Section: Overview of ALS and FTD genes. The text in this section has been revised.

Most of the heritability of FTD is accounted for by autosomal dominant mutations in three genes: progranulin (GRN), microtubule-associated protein tau (MAPT), and C9orf72 (129). In recent years, an increasing number of mutations in other genes have been associated with autosomal dominant FTD, e.g., VCP (2004), CHMP2B (2005), TARDBP (2008), FUS (2009), SQSTM1 (2012), CHCHD10 (2014), TBK1 (2015), OPTN (2015), CCNF (2016), TIA1 (2017). Recent studies have identified TBK1 as probably the fourth most common genetic cause overall of FTD, accounting for between 1% and 2% of all cases (although the pathogenic nature of many of the reported missense variants remains unclear) (130).

SOD1: the same as comment 2.

  1. The focus of the review on describing the overlap in the genetic background of ALS and FTD gets often lost and entire subsections do not even mention FTD. As an example, mutation in SOD1 are known to associate with pure motoneuronal disease, and yet ample space is dedicated to that.

We agree again with the reviewer; this point merits some concern.

In most families harboring SOD1 gene mutations, disease penetrance is > 90% by age 70 yrs. (141), and more than 170 mutations have now been detected in the fALS SOD1 gene (142)

Pure upper motor neuron (UMN) and lower motor neuron (LMN) forms have also been described representing opposite clinical ends of the MND spectrum (9). Mutations in the SOD1 gene could be associated with significant LMN involvement with or without signs of UMN. The A4V missense mutation occurs in around 40% of patients in North America but is rare in the European population. LMN signs predominate, with features of UMN being mild or absent. Disease progression is particularly rapid, with a median survival of 1.2 years from disease onset (151). The A4T mutation is also associated with a similarly rapid disease course for LMN predominant syndrome (152). In contrast, the G93C mutation has been associated with a pure clinical phenotype of LMN, i.e.,  without bulbar involvement and a more favorable prognosis (i.e., a median survival of 153 months) (153).

  1. Similarly, in the “Mitochondria dysfunction” section FTD is not even considered.

Thank you for this comment. We modified these sections accordingly, as follows.

Mitochondrial DNA instability disorders are responsible for the broad clinical spectrum of ALS and FTD.

Mitochondrial DNA instability disorders are responsible for frontotemporal dementia (68). In recent years, a growing list of FTD genes responsible for mitochondrial DNA instability has been reported (70). The c.176C>T mutation in the CHCHD10 gene was described in an FTD-ALS patient whose family was originally from Catalonia (Spain), with affected individuals carrying a missense mutation in the CHCHD10 gene. Functional characterization of the CHCHD10 mutant identified in the family showed fragmentation of the mitochondrial network and the loss of cristae junctions (68). CHCHD10 is a novel gene responsible for the clinical spectrum of ALS-FTD, which raises the intriguing prospect of an underlying mitochondrial basis for this group of disorders.

  1. The section of RNA metabolism is very short, and the authors neglect to mention any of the many studies that have investigated RNA dysregulation in ALS and FTD using human tissue, animal, and cellular disease models. Only two broad-scoped reviews are referenced.

Thank you for this suggestion. We suggest the following text modification:

As key regulators of RNA metabolism, RNA-binding proteins (RBP) play a critical role in maintaining the normal function of neuronal systems. RNA-binding proteins are involved in several aspects of RNA metabolism, including splicing, transcription, transport, translation, and storage in stress granules (91). The aggregation of RBP is a pathological hallmark of amyotrophic lateral sclerosis and frontotemporal lobar degeneration.   

Mutations in ALS genes contribute to the etiology of FTD and vice versa (10). Many ALS-causing mutations impact proteins involved in RNA metabolism, including RNA-binding proteins such as TDP-43, FUS, and heterogeneous nuclear ribonucleoprotein A1 (hnRNPA1) (93). These and related RNA-binding proteins are components of organelles without membranes found in the nucleus (e.g., nuclear speckles and nucleoli) and cytoplasm (e.g., processing bodies and stress granules) in neurons and other cell types (94) (95) (96).

Mechanisms Leading to Dysregulation of RBP in ALS

Mutations in genes encoding many RBP are highly associated with ALS. In addition, dysregulation of RBP as a result of compromised nucleocytoplasmic trafficking, posttranslational modification (PTM), aggregation, and sequestration by abnormal RNAs also contributes significantly to disease pathogenesis. This section will briefly discuss the underlying mechanisms resulting in RBP dysregulation in ALS (97).

In response to a variety of stressors such as heat shock and oxidative insult, TDP-43 and FUS translocate from the nucleus and associate with cytoplasmic stress granules (SG), which are dense aggregations of protein-RNA complexes (98) (99). RBPs recruited to stress granules under conditions of chronic stress are capable of forming insoluble protein aggregates, even when other components of the stress granules have dissociated from the complex (100).

RBP also contains low complexity sequence domains (LCD), i.e., a glycine-rich domain that promotes protein aggregation (102) and contains RNA-recognition motifs (RRM) necessary for the nucleic acid binding functions of the protein (103). Each protein also contains a nuclear localization sequence (NLS) that directs the subcellular localization of the protein to the nucleus under normal conditions (104). Mutations in genes encoding NLS and LCD (Fig. 1) lead to cytoplasmic retention and inclusion formation in cultured cells (105). More than 250 proteins with aggregation-prone properties that are likely to contribute to neurodegeneration have been identified (106).

Figure 1. Structure of RBP genes

Low Complexity sequence Domains (LCD), Nuclear Localization Sequence (NLS).  Mutations occur in these domains (LCD and NLS) can trigger the same pathological cascade which leads to a deterioration in the dynamics of stress granules (updated following original citation Baradaran-Heravi et al, 2019).

These studies suggest that such RBP could or should be considered as potential functional candidate genes in genetic studies. New RBM45, an RNA-binding protein, is the most likely causal gene for ALS-FTD. In addition, a novel and evolutionary conserved structural element homo-oligomer assembly (HOA) domain has been identified. It is located within the linker region between RNA-recognition motifs (RRM2 and RRM3), which are essential for the self-association and oligomerization of RBM45 (Fig. 2). Since RBM45 contains three RRM domains, it may associate with TDP-43 and FUS through RNA-protein interactions (107).

Figure 2. Structure RBM45 gene

RNA-recognition motifs (RRM), homo-oligomer assembly (HOA). RBM45 lacks the typical low complexity domain (LCD), which is actually common in RBPs; it has a suspicious homo oligomerization domain that, similar to LCD, mediates self-assembly through homo oligomerization and its interaction with other proteins (updated following original citation Li et al. 2015).

Another RBP is TIA-1 that promotes the assembly of stress granules discrete cytoplasmic inclusions into which stalled translation initiation complexes are dynamically recruited in cells subjected to environmental stress (108). TIA-1 is a modular protein composed of three RNA recognition motifs and a carboxy-terminal glutamine-rich motif that is structurally related to prion protein (PRD). Overexpressed TIA-1 induces SG formation and represses reporter gene expression, whereas the isolated prion-related domain (PRD) of TIA-1 forms cytoplasmic microaggregates (109). These data suggest that the PRD is capable of self-oligomerization, just like RBM45.

  1. Some statements are incorrect, such as the fact that C9ORF72 protein forms cytoplasmic aggregates (page 3, line 104), and citations imprecise (e.g. Ref#29 on line 179).

Thanks for pointing this out; we appreciate it.

The accumulation of (GGGGCC) might lead to sequestration of RNA binding proteins and disruption of the translation of diverse mRNA or increased nucleolar stress (46).

Dafinca R, Scaber J, Ababneh N, Lalic T, Weir G, Christian H, Vowles J, Douglas AG, Fletcher-Jones A, Browne C, Nakanishi M, Turner MR, Wade-Martins R, Cowley SA, Talbot K. C9orf72 Hexanucleotide Expansions Are Associated with Altered Endoplasmic Reticulum.

  1. Overall, a major rework of both the content and structure of this review would be required to make it acceptable for publication.

We thank the reviewer for this helpful comment; this confusion was made by using a different citation manager. We corrected and updated the references throughout the manuscript.

Reviewer 2 Report

Summary of Paper:

            The authors proved a detailed review of current understanding within ALS and FTLD and their similarities in genetic background. This reviewer sees no reason to delay publication of this manuscript; however, there are a few minor points that the authors should consider before final submission.

Minor Issues:

  • Grammatical errors. There are several minor grammatical errors that the authors should clean up before submission. For example, on lines 103-104 there is no need for parentheses after “including (RNA foci, etc)”.

This reviewer recommends the authors insert a few figures or tables within the manuscript to make it easier to follow. Two places that would be helpful would be lines 115 – 126 and 162-174

Author Response

The authors proved a detailed review of current understanding within ALS and FTLD and their similarities in genetic background. This reviewer sees no reason to delay publication of this manuscript; however, there are a few minor points that the authors should consider before final submission.

Minor Issues:

  1. Grammatical errors. There are several minor grammatical errors that the authors should clean up before submission. For example, on lines 103-104 there is no need for parentheses after including (RNA foci, etc.).

Thanks for the alert. Grammatical errors checked and corrected. The text has been rewritten. The accumulation of (GGGGCC) might lead to sequestration of RNA binding proteins and disruption of the translation of diverse mRNA or increased nucleolar stress (46).

  1. This reviewer recommends the authors insert a few figures or tables within the manuscript to make it easier to follow. Two places that would be helpful would be lines 115 – 126 and 162-174.

Thank you for this comment. Figures have been inserted that graphically show the individual structures of RBP, and a table was added with new genes involved in ASL-FTD.

Figure 1. Structure of RBP genes

Figure 2. Structure of the RBM45 gene

Figure 3. ALS-related genes

Gene

Locus

Inheritance

Mutation freguency

Protein function

fALS

%

sALS

%

Overal ALS %

fTDP

%

sTDP

%

Overal TDP %

TBK1

12q14.2

AD

3

<1

1,3

2

1

<1

autophagy

CHCHD

22q11.23

AD

2

<1

<1

<1

<1

<1

mitochondrial dysfunction, synaptic integrity

TIA1

2p13.3

AD

2

<1

RNA metabolism

ANXA11

10q22.3

AD

1,2

<1

1,1

apoptosis, exocytosis, cytokinesis

CCNF

16p13.3

AD

0,6-3,3

<1

<1

4

UPS

NEK1

4q33

?

1-2

<1

DNA damage, mitochondrial membrane regulation

C21orf2

21q22.3

?

<1

<1

<1

DNA damage

MATR3

5q31.2

AD

1

1

<1

RNA metabolism

TUBA4A

2q35

AD

1

<1

<1

<1

<1

cytoskeletal dynamics

Table 1. Overview of recent ALS genes in different ALS and TDP

AD, autosomal dominant, UPS, ubiquitun-proteasome system

Round 2

Reviewer 1 Report

The authors have done a good job of addressing my previous concerns. A few minor issues remaining are listed below:

  1. On page 3, line 125 - the sentence “Mitochondrial DNA instability disorders are responsible for the broad clinical spectrum of ALS and FTD” should be modified to something more broadly accepted, such as “”mitochondria dysfunction is at least partly responsible for ....”
  2. The new section added on page 3, lines 148-157 is still redundant. The authors could consider incorporating it better with the rest of the paragraph.
  3. Page 5 line 258 - the sentence “ New RBM45, an RNA-binding protein, is the most likely causal gene for ALS-FTD.” should be rephrased “ RBM45, an RNA-binding protein, is most likely a causal gene for ALS-FTD.”
  4. Page 6 line - the sentence “RNA-recognition motifs (RRM), homo-oligomer assembly (HOA).” has no meaning and should be revised.

Author Response

Our manuscript has been revised twice by a native speaker, if you request another revision of the English language as required by the journal, we will pay for it.
